# Identifying areas prone to coastal hypoxia - the role of topography

Elina A. Virtanen[1,2], Alf Norkko[3,4], Antonia Nyström Sandman[5], Markku Viitasalo[1]

[1]Marine Research Centre, Finnish Environment Institute, Helsinki, 00790, Finland
[2]Department of Geosciences and Geography, University of Helsinki, Helsinki, 00014, Finland
[3]Tvärminne Zoological Station, University of Helsinki, Hanko, 10900, Finland
[4]Baltic Sea Centre, Stockholm University, Stockholm, 10691, Sweden
[5]AquaBiota Water Research, Stockholm, 11550, Sweden

*Correspondence to*: Elina A. Virtanen (elina.a.virtanen@environment.fi)

**Abstract.** Hypoxia is an increasing problem in marine ecosystems around the world. While major advances have been made in our understanding of the drivers of hypoxia, challenges remain in describing oxygen dynamics in coastal regions. The complexity of many coastal areas and lack of detailed *in situ* data has hindered the development of models describing oxygen dynamics at a sufficient spatial resolution for efficient management actions to take place. It is well known that the enclosed nature of seafloors and reduced water mixing facilitates hypoxia formation, but the degree to which topography contributes to hypoxia formation, and small-scale variability of coastal hypoxia, has not been previously quantified. We developed simple proxies of seafloor heterogeneity and modelled oxygen deficiency in complex coastal areas in the northern Baltic Sea. According to our models, topographical parameters alone explained ~80 % of hypoxia occurrences. The models also revealed that less than 25 % of the studied seascapes were prone to hypoxia during late summer (August-September). However, large variation existed in the spatial and temporal patterns of hypoxia, as certain areas were prone to occasional severe hypoxia ($O_2 < 2$ mg L$^{-1}$), while others were more susceptible to recurrent moderate hypoxia ($O_2 < 4.6$ mg L$^{-1}$). Areas identified as problematic in our study were characterized by low exposure to wave forcing, by high topographical shelter from surrounding areas, and by isolation from the open sea, all contributing to longer water residence times in seabed depressions. Deviations from this "topographical background" are probably caused by strong currents or by high nutrient loading, thus improving or worsening oxygen status, respectively. In some areas, connectivity with adjacent deeper basins may also influence coastal oxygen dynamics. Developed models could boost the performance of biogeochemical models, aid developing nutrient abatement measures, and pinpoint areas where management actions are most urgently needed.

## 1 Introduction

Hypoxia is a key stressor of marine environments, occurring in over 400 physically diverse marine ecosystems worldwide (Diaz and Rosenberg, 1995b, 2008;Conley et al., 2009b). Declining oxygen levels have been recorded in fjords, estuaries and in coastal and open-sea areas, such as Chesapeake Bay, Gulf of Mexico, Japan Sea, Baltic Sea and the Black Sea (Gilbert et al., 2010;Carstensen et al., 2014). It is clear that our oceans are losing their breath, and recent projections indicate that anoxic zones devoid of higher life will be increasing in the forthcoming decades (Frölicher et al., 2009;Meier et al., 2011a;Meier et al., 2012a), with severe consequences for marine ecosystems (Breitburg et al., 2018).

The lack of oxygen alters the structure and functioning of benthic communities (Nilsson and Rosenberg, 2000;Gray et al., 2002;Karlson et al., 2002;Valanko et al., 2015), disrupts bioturbation activities (Timmermann et al., 2012;Villnas et al., 2012;Villnas et al., 2013;Norkko et al., 2015), changes predator-prey relationships (Eriksson et al., 2005) and may lead to mass mortalities of benthic animals (Vaquer-Sunyer and Duarte, 2008). Hypoxia does not only affect organisms of the seafloor, but also influences biogeochemical cycling and benthic-pelagic coupling (Gammal et al., 2017). Hypoxia can increase releases of nutrients from the sediment and thus promote planktonic primary production and sedimentation, which in turn leads to enhanced microbial consumption of oxygen (Conley et al., 2002;Kemp et al., 2009;Middelburg and Levin, 2009). This creates a self-sustaining process, often referred to as "vicious circle of eutrophication" (Vahtera et al., 2007), which may hamper the effects of nutrient abatement measures.

Biogeochemical processes contributing to hypoxia formation are well known. Factors affecting the development of hypoxia are usually associated with the production of organic matter, level of microbial activity and physical conditions creating stratification and limited exchange or mixing of water masses (Conley et al., 2009a;Rabalais et al., 2010;Conley et al., 2011;Fennel and Testa, 2019). Coastal hypoxia is common in areas with moderate or high anthropogenic nutrient loading, high primary productivity and complex seabed topography limiting lateral movement of the water. Shallow-water hypoxia is often seasonal. It is associated with warming water temperatures and enhanced microbial processes and oxygen demand (Buzzelli et al., 2002;Conley et al., 2011;Caballero-Alfonso et al., 2015;van Helmond et al., 2017).

Projecting patterns and spatial and temporal variability of hypoxia is necessary for developing effective management actions. Thus three-dimensional coupled hydrodynamic-biogeochemical models have been created for several sea areas around the world, such as Gulf of Mexico (Fennel et al., 2011;Fennel et al., 2016), Chesapeake Bay (Scully, 2013;Testa et al., 2014;Scully, 2016), the North Sea (Hordoir, 2018) and the Baltic Sea (Eilola et al., 2009;Eilola et al., 2011;Meier et al., 2011a;Meier et al., 2012a;Meier et al., 2012b). These models simulate various oceanographic, biogeochemical and biological processes using atmospheric and climatic forcing and information on nutrient loading from rivers. While such models are useful for studying processes at the scale of kilometers, and aid in defining hypoxia abatement at the basin-scale, their horizontal resolution is too coarse (often 1−2 nautical miles) for accurately describing processes in coastal areas. Lack of detailed data on water depth, currents, nutrient loads, stratification and local distribution of freshwater discharges (Breitburg et al., 2018) (not to mention computational limitations) usually prevent the application of biogeochemical models developed

to large geographical areas at finer horizontal resolutions (<100 m). Understanding spatial variability of hypoxia in
topographically complex coastal environments has therefore been impeded by the lack of useful methods and systematic,
good-quality data (Diaz and Rosenberg, 2008;Rabalais et al., 2010;Stramma et al., 2012). Finding alternative ways to
pinpoint areas prone to coastal hypoxia could facilitate management and determining of efficient local eutrophication
abatement measures.
It is widely recognized that the semi-enclosed nature of the seafloors, and associated limited water exchange is a significant
factor in the formation of hypoxia in coastal waters (Diaz and Rosenberg, 1995a;Virtasalo et al., 2005;Rabalais et al.,
2010;Conley et al., 2011).  However, to determine the degree to which seascape structure restricting water movement
contributes to hypoxia formation has not been quantified. Analytical and theoretical frameworks developed specifically for
terrestrial environments, such as landscape heterogeneity or patchiness, are analogous in marine environments, and are
equally useful for evaluating links between ecological functions and spatial patterns in marine context. We tested how large
fraction of hypoxia occurrences could be explained only by structural complexity of seascapes, without knowledge on
hydrographical or biogeochemical parameters. We adopted techniques and metrics from landscape ecology and transferred
them to marine environment, and (1) examined if spatial patterns in seascapes can explain the distribution of hypoxia, (2)
defined the relative contribution of seascape structure to hypoxia formation and (3) estimated the potential ranges of hypoxic
seafloors in coastal areas. To achieve this, we concentrated on extremely heterogeneous and complex archipelago areas in
the northern Baltic Sea, where coastal hypoxia is a common and an increasing problem (Conley et al., 2011;Caballero-
Alfonso et al., 2015).
**2 Data and methods**
**2.1 Study area**
The studied area covers the central northern Baltic Sea coastal rim, 23 500 km$^2$ of Finnish territorial waters from the
Bothnian Bay to the eastern Gulf of Finland, and 5100 km$^2$ of Swedish territorial waters in the Stockholm Archipelago in the
Baltic Proper. Oxygen dynamics in the deeper areas of the Gulf of Finland and Stockholm Archipelago are strongly affected
by oceanography and biogeochemistry of the central Baltic Proper, not reflecting the dynamics of coastal hypoxia (Laine et
al., 1997), and were therefore excluded from this study. The outer archipelago of Finland is relatively exposed with various
sediment and bottom habitat types, while the inner archipelago is more complex and shallower, but maintains a higher
diversity of benthic habitats and sediment types (Valanko et al., 2015). The inner archipelago of Stockholm is an equally
complex archipelago area, with a large number of islands, straits and coves. Freshwater outflow from Lake Mälaren creates
an estuarine environment where freshwater meets the more saline water in the Baltic Proper.
In order to evaluate differences in oxygen deficiency between coastal areas, the study area was divided into five regions as
defined by the EU Water Framework Directive (2000/60/EC) (WFD, 2000): the Archipelago Sea (AS), the Eastern Gulf of
Finland (EGoF), the Gulf of Bothnia (GoB), Stockholm Archipelago (SA) and the western Gulf of Finland (WGoF) (Figure
1). Small skerries and sheltered bays characterize AS, EGoF and WGoF, whereas narrower band of islands forms relatively
exposed shores in GoB. Deep, elongated channels of bedrock fractures can reach depths of over 100 m in AS, similarly to
SA where narrow, deep valleys separate mosaics of islands and reefs. Substrate in both areas varies from organic-rich soft
sediments in sheltered locations to hard clay, till and bedrock in exposed areas. At greater depths, soft sediments are
common due to limited water movement. As a whole, the study area with rich topographic heterogeneity forms one of the
most diverse seabed areas in the world (Kaskela et al., 2012). In many areas hypoxia is a result of strong water stratification,
slow water exchange and complex seabed topography, creating pockets of stagnant water (Conley et al., 2011;Valanko et al.,
2015;Jokinen et al., 2018). Thus, the area is ideal for testing hypotheses of topographical controls for hypoxia formation.

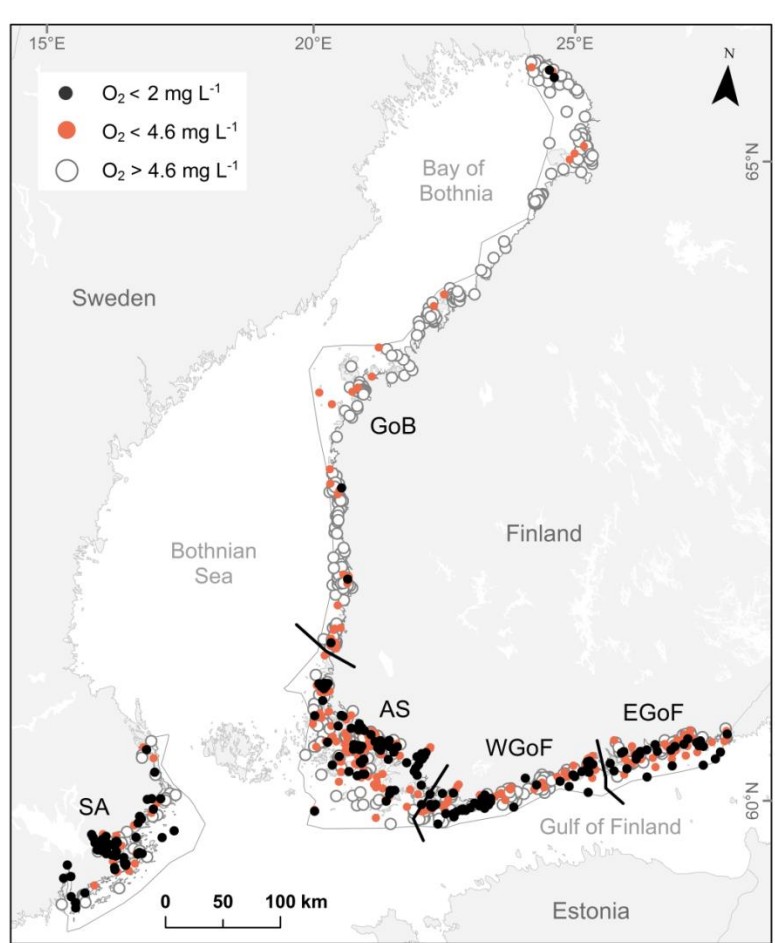

**Figure 1**. Study areas in Finland: AS – Archipelago Sea, WGoF – Western Gulf of Finland, EGoF – Eastern Gulf of Finland,
GoB – Gulf of Bothnia, and Sweden: SA – Stockholm Archipelago. Orange dots represent sites of moderately hypoxic ($O_2 <$
4.6 mg L -1) black dots severely hypoxic ($O_2 <$ 2 mg L -1) and white circles denote sites with $O_2 >$ 4.6 mg L -1. Grey and
black lines illustrate boundaries of Water Framework Directive areas.

## 2.2 Hypoxia data

Bottom-water hypoxia is the main factor structuring benthic communities in the Baltic Sea (Villnas et al., 2012;Norkko et al., 2015). Two mg L$^{-1}$ of $O_2$ is usually considered a threshold where coastal organisms start to show symptoms of the lack of oxygen, and this limit has been commonly used in various global reviews (Diaz and Rosenberg, 1995b, 2008). Some studies have however concluded that 2 mg L$^{-1}$ is below the empirical sublethal and lethal oxygen limit for many species (Vaquer-Sunyer and Duarte, 2008;Conley et al., 2009b). Here we define hypoxia based on two ecologically meaningful limits: moderately hypoxic <4.6 mg L$^{-1}$ $O_2$ – as this has been estimated to be a minimum safe limit for species survival, behavior and functioning in benthic communities (Norkko et al., 2015) – and severely hypoxic <2 mg L$^{-1}$ $O_2$, which describes zones where larger marine organisms suffer from severe mortality (Vaquer-Sunyer and Duarte, 2008). As no reference values exist for severity of hypoxia to marine organisms based on the frequency of hypoxic events (Norkko et al., 2012;Villnas et al., 2012;Norkko et al., 2015), we here define a site to be prone to hypoxic events, i.e. occasionally hypoxic, if it experienced hypoxia ($O_2$<2 mg L$^{-1}$ and <4.6 mg L$^{-1}$) at least once during the study period. If hypoxia was recorded ≥20 % of the visits, it was categorized as frequently hypoxic. We consider this to be ecologically relevant, as species develop symptoms already from short exposures to hypoxia (Villnas et al., 2012;Norkko et al., 2015). This is also justified, as our oxygen data is from ~1 m above the seafloor, suggesting that the actual oxygen concentrations at sediment where benthic species live are probably lower.

Data from oxygen profiles were collated from the national monitoring environmental data portals Hertta (http://www.syke.fi/en-US/Open_information) and SHARK (https://www.smhi.se/data/oceanografi/havsmiljodata). Data was available from 808 monitoring sites. Only months of August and September 2000–2016 were considered, as hypoxia is usually a seasonal phenomenon occurring in late summer when water temperatures are warmest (Conley et al., 2011).

## 2.3 Predictors

For modelling hypoxia occurrences, we developed five geomorphological metrics: (1) Bathymetric Position Indices (BPI) with varying search radii, (2) Depth-Attenuated Wave Exposure (SWM(d)), (3) Topographical Shelter Index (TSI), (4) Arc-Chord Rugosity (ACR) and (5) Vector Ruggedness Measure (VRM). BPI is a marine modification of the terrestrial version Topographic Position Index (TPI), originally developed for terrestrial watersheds (Weiss, 2001). BPI is a measure of a bathymetric surface to be higher (positive values) or lower (negative values) than the overall seascape. BPI values close to zero are either flat areas or areas with constant slope. Here, BPIs represent topographical depressions and crests at scales of 0.1, 0.3, 0.5, 0.8, and 2 km calculated with Benthic Terrain Modeler (v3.0) (Walbridge et al., 2018). SWM(d) estimates dominant wave frequency at a given location with the decay of wave exposure with depth, and takes into account diffraction and refraction of waves around islands (Bekkby et al., 2008). SWM(d) characterizes areas where water movement is slower, i.e. where water resides longer. In terrestrial realms, landform influence on windthrow patterns, i.e. exposure to winds, have been noted in several studies, e.g. Kramer (2001) and Ashcroft et al. (2008). Here, we introduce an analogous version of

"windthrown-prone" areas to the marine realm, i.e. a "wave-prone" metric: Topographical Shelter Index (TSI), which
differentiates wave directions and takes into account the sheltering effects of islands (i.e. exposure above sea-level).
Identification of "wave-prone" areas was calculated for the azimuths multiple of 15 (0°–345°), and for altitudes
(corresponding to the angle of light source) ranging from 0.125° to 81°. For each altitude the produced "wave-prone areas"
were combined to an index value for each grid cell. Surface roughness is a commonly used measure of topographical
complexity in terrestrial studies, and has been used in marine realms as well (see e.g. Dunn and Halpin (2009) for modelling
habitats of hard substrates and Walker et al. (2009) for complexity of coral reef habitats). Here, we consider two approaches
for estimating seascape rugosity; Arc-Chord Rugosity (ACR) and Vector Ruggedness Measure (VRM). ACR is a landscape
metrics, which evaluates surface ruggedness using a ratio of contoured area (surface area) to the area of a plane of best fit,
which is a function of the boundary data (Du Preez, 2015). VRM on the other hand, is a more conservative measure of
surface roughness developed for wildlife habitat models, and is calculated using a moving 3×3 window where a unit vector
orthogonal to the cell is decomposed using the three dimensional location of the cell center, the local slope and aspect. A
resultant vector is divided by the number of cells in the moving window (Sappington et al., 2007). Both rugosity indices
were used here to identify areas with complex marine geomorphology. Differences of predictor variables are illustrated in
Fig. 2. We also included geographical study areas as predictors, in order to highlight the differences between WFD areas.

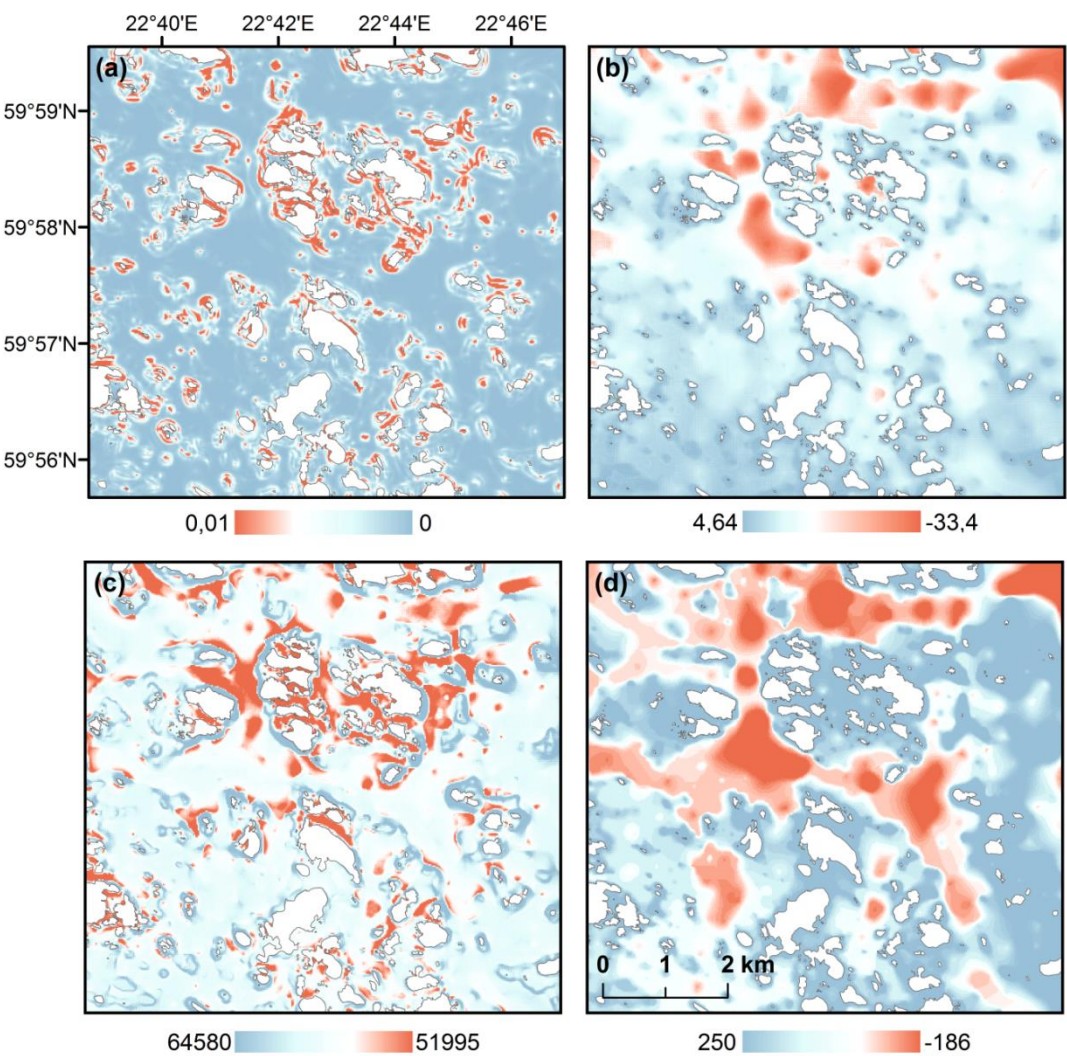

**Figure 2.** Predictor variables developed for hypoxia ensemble models: a) Vector Ruggedness Measure (VRM), b) Depth-Attenuated Wave Exposure (SWM(d)), c) Topographical Shelter Index (TSI) and (d) Bathymetric Position Index (BPI) with a search radius of 2 km. Red color represents rugged seafloors (VRM), sheltered areas (SWM(d), TSI) and depressions (BPI2). Islands shown as white.

**2.4 Hypoxia models**

Based on the ecologically meaningful limits of hypoxia (see section "Hypoxia data"), we built four separate oxygen models based on frequency and severity of hypoxia: occasional with $O_2$ limits <4.6 and <2 mg $L^{-1}$ (hereafter referred as $OH_{4.6}$ and $OH_2$) and frequent hypoxia with $O_2$ limits <4.6 and <2 mg $L^{-1}$ (hereafter referred as $FH_{4.6}$ and $FH_2$). We used Generalize Boosted Regression Models (GBM) and its extension Boosted Regression Trees (BRT), a method from statistical and

machine learning traditions (De'ath and Fabricius, 2000;Hastie et al., 2001;Schapire, 2003). BRT optimizes predictive
performance through integrated stochastic gradient boosting (Natekin and Knoll, 2013), and forms the best model for
prediction based on several models.
Ideal model tuning parameters (learning rate, bag fraction, tree complexity) for our hypoxia models were based on
optimizing the model performance, i.e. minimizing the prediction error. The learning rate was set to 0.001 to determine the
contribution of each successive tree to the final model. We varied the number of decision rules controlling model interaction
levels, i.e. tree complexities, between 3 and 6. We shuffled the hypoxia data randomly into ten subsets for training (70 %)
and testing (30 %), while preserving the prevalence ratio of hypoxia occurrence. Hypoxia models were developed based on
10-fold cross-validation, and resulting final, best models leading to smallest predictive errors were chosen to predict the
probability of detecting hypoxia across the study region at a resolution of 20 m. We believe such a high resolution is
necessary due to the complexity of archipelagoes of Finland and Sweden. Model predictions for the whole seascape were
repeated ten times with different model fits from data subsets (40 separate model predictions), to identify areas where
models agree on the area to be potentially hypoxic. Model performances were estimated against the independent data (test
data 30 %), not used in model fitting in order to evaluate the potential overfitting of the models. Analyses were performed in
R 3.5.0. (R, 2018) with R libraries 'gbm' (Greenwell et al., 2018), 'PresenceAbsence' (Freeman and Moisen, 2008) and
relevant functions from Elith et al. (2008).
Variable selection in BRT is internal by including only relevant predictors when building models. The importance of
predictors is based on the time each predictor is chosen in each split, averaged over all trees. Higher scores (summed up to
100 %) indicate that a predictor has a strong influence on the response (Elith et al., 2008). Although BRT is not sensitive to
collinearity of predictors, the ability to identify strongest predictors by decreasing the estimated importance score of highly
correlated ones detriments the interpretability of models (Gregorutti et al., 2017). Selecting only optimal and a minimal set
of variables for modelling, i.e. finding all relevant predictors and keeping the number of predictors as small as possible,
reduces the risk of overfitting and improves the model accuracy. Here, most of the predictors describe seascape structure and
are somehow related to the topography of the seabed. We estimated the potential to drop redundant predictors, i.e. those that
would lead to marked improvement in model performance if left out from the model building. For this, we used internal
backward feature selection in BRT. However, we did not find marked differences in predictive performances, and used all
predictors.
Estimation of model fits and predictive performances was based on the ability to discriminate a hypoxic site from an oxic
one, evaluated with Area Under the Curve (AUC) (Jiménez-Valverde and Lobo, 2007) and simply with Percent Correctly
Classified (PCC) (Freeman and Moisen, 2008). AUC is a measure of detection accuracy of true positives (sensitivity) and
true negatives (specificity), and AUC values above 0.9 indicate excellent, 0.7–0.9 good, and below 0.7 poor predictions.
We transformed hypoxia probability predictions into binary classes of presence/absence, and estimated the relative area of
potentially hypoxic due to topographical reasons. Although dichotomization of probability predictions flattens the
information content, it facilitates the interpretation of results, and is needed for management purposes. Predicted range of

hypoxia and the potential geographical extent enables the identification of problematic areas and facilitates management actions in a cost-effective way. There are various approaches for determining thresholds, which are based on the confusion matrix, i.e. how well the model captures true/false presences or true/false absences. Usually the threshold is defined to maximize the agreement between observed and predicted distributions. Widely used thresholds, such as 0.5, can be arbitrary unless the threshold equals prevalence of presences in the data, i.e. the frequency of occurrences (how many presences of the total dataset) (Liu et al., 2005). Here, we define thresholds objectively based on an agreement between predicted and observed hypoxia prevalence. This approach underestimates areas potentially hypoxic (see section "Hypoxia areas") and is expressed here as a conservative estimate.

## 3 RESULTS

### 3.1 Hypoxia in complex coastal archipelagos

During 2000–2016 hypoxia was rather common throughout the whole study region. In Finland, hypoxia mostly occurred on the southern coast, as in Archipelago Sea (AS), Eastern Gulf of Finland (EGoF) and Western Gulf of Finland (WGoF) hypoxic events were recorded frequently. Only ~30 % of coastal monitoring sites in AS, EGoF and SA were not hypoxic, i.e., oxygen concentrations were always above 4.6 mg L$^{-1}$ (cf. percentages in brackets in Fig. 3). Coastal areas in Stockholm Archipelago (SA) were also regularly hypoxic, with 70 % of the sites moderately ($O_2 < 4.6$ mg L$^{-1}$) and 53 % severely ($O_2 < 2$ mg L$^{-1}$) hypoxic. Severe hypoxia was quite a localized phenomenon in Finland, as it was recorded at ca. 30 % of sites in AS and EGoF. However, in WGoF there are sites where severe hypoxia is rather persistent, as every sampling event was recorded as hypoxic. Same applies to SA, as there are quite a few sites repeatedly severely hypoxic. In contrast in the northern study area, Gulf of Bothnia (GoB), hypoxic events occurred rather infrequently, as in 98 % of sites $O_2$ was above 2 mg L$^{-1}$ and in 87 % $O_2$ was above 4.6 mg L$^{-1}$ (Fig. 3).

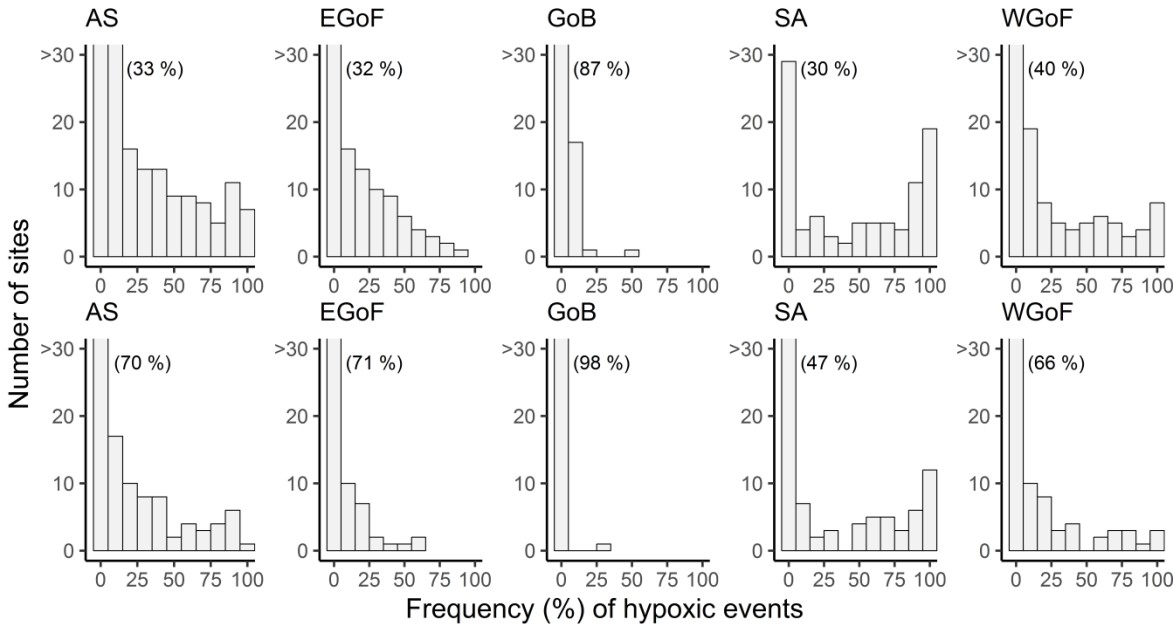

**Figure 3**. Frequencies of hypoxic events at coastal monitoring sites across Water Framework Directive areas: Archipelago Sea (AS), Eastern Gulf of Finland (EGoF), Gulf of Bothnia (GoB), Stockholm Archipelago (SA) and Western Gulf of Finland (WGoF). Upper panels indicate $O_2 < 4.6$ mg L$^{-1}$, lower panels $O_2 < 2$ mg L$^{-1}$. Numbers in brackets indicate the percentage of sites with $O_2 > 4.6$ (upper panel) and $O_2 > 2$ mg L$^{-1}$ (lower panel). Sites >30 are not shown.

## 3.2 Importance of predictors

Models developed on 566 sites, based on 10-fold cross-validation and ten repeated predictions, the most influential predictor (averaged across models) was SWM(d), with a mean contribution of 33 % (± 5 %) (Fig. 4). The contribution of SWM(d) was highest for frequent, severe hypoxia (FH$_2$) (41 ± 3 %), whereas for occasional, moderate hypoxia (OH$_{4.6}$) the influence was markedly lower (25 ± 5 %). This supports the hypothesis that in sheltered areas, where water movement is limited, severe oxygen deficiency is likely to develop. Noteworthy is also that depth was not the most important driver of hypoxia in coastal areas. This suggests that coastal hypoxia is not directly dependent on depth, but that depressions that are especially steep and isolated are more sheltered and become more easily hypoxic than smoother depressions.

Across models, BPIs identifying wider sinks (BPI2 and BPI0.8) were more influential than BPIs identifying smaller sinks (BPIs 0.1, 0.3 and 0.5), and terrain ruggedness measures, VRM and ACR, were more important for frequent severe hypoxia (FH$_2$) than for moderate hypoxia (FH$_{4.6}$). The relatively high contribution of topographical shelter (TSI 7 ± 2 %) indicates that, in areas where there are higher islands, the basins between are prone to hypoxia formation.

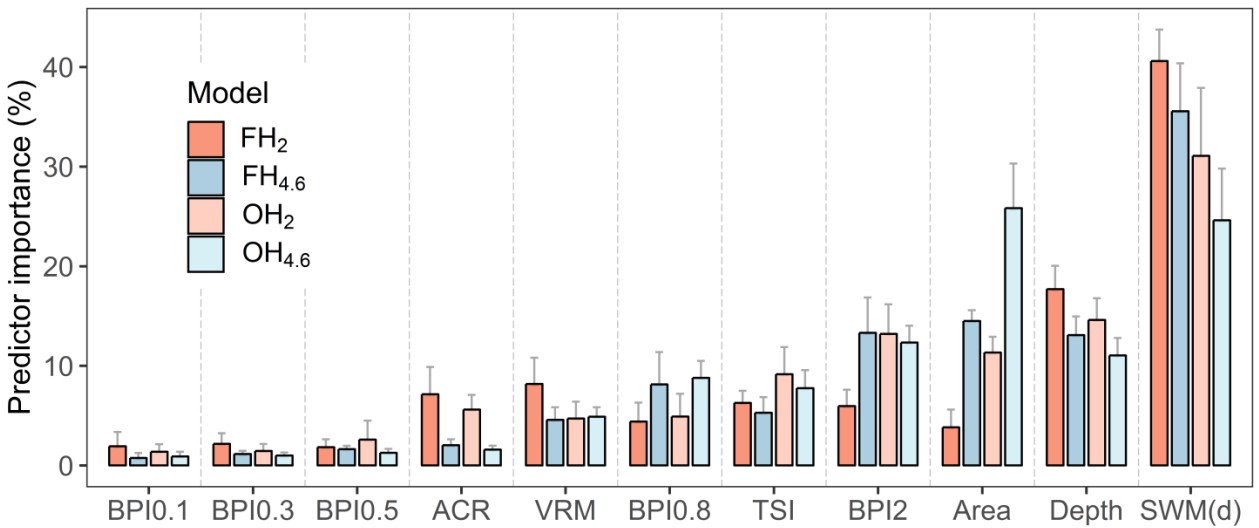

**Figure 4.** Importance of predictors based on ten prediction rounds. Predictors are colour-coded based on models. FH$_2$= frequent, severe hypoxia (O$_2$ < 2 mg L$^{-1}$), FH$_{4.6}$= frequent, moderate hypoxia (O$_2$ < 4.6 mg L$^{-1}$), OH$_2$= occasional, severe hypoxia (O$_2$ < 2 mg L$^{-1}$), and OH$_{4.6}$= occasional, moderate hypoxia (O$_2$ < 4.6 mg L$^{-1}$). Whiskers represent standard deviations.

## 3.3 Model performance

Predictive ability of models to detect sites as hypoxic across models was good, with a mean 10-fold cross-validated AUC of 0.85 (± 0.02) and mean AUC of 0.86 (± 0.03) when evaluated against independent test data for 242 sites (30 % of sites) (Fig. 5a). Models classified on average 88 % sites correctly (PCCcv in Fig. 5b), and performed only slightly worse when evaluated against independent data, with 81 % (± 3 %) correctly classified (PCCin in Fig. 5b). Models developed for frequent hypoxia ($FH_2$ and $FH_{4.6}$) were better (mean AUCin 0.88 ± 0.03) compared to occasional hypoxia models (mean AUCin 0.84 ± 0.04). This suggests that other factors beyond topographical proxies contribute relatively more to the occurrence of occasional hypoxia than for frequent hypoxia.

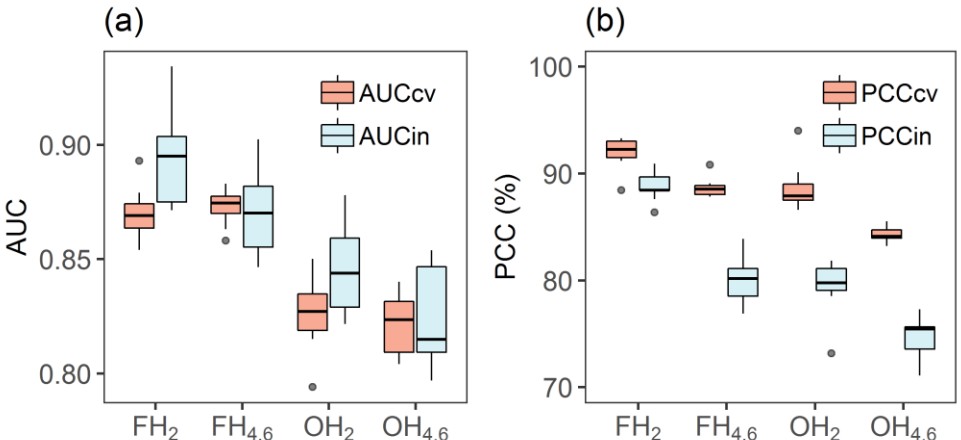

**Figure 5.** Model performances based on (a) Area Under the Curve values with 10-fold cross-validation (AUCcv), and against independent test data (30 % of sites) (AUCin), (b) Percent of Correctly Classified with 10-fold cross-validation (PCCcv) and against independent data (PCCin).

### 3.4 Hypoxic areas

Although hypoxia was commonly recorded in all WFD areas, except in the Gulf of Bothnia, the potential geographical extent of hypoxic seafloors shows rather different pattern. Based on models, topographically prone areas represent only a small part of the coastal areas, with less than 25 % affected (Fig. 6). Frequent, severe hypoxia ($O_2 < 2$ mg L$^{-1}$) was most prominent in the Archipelago Sea and Stockholm Archipelago, although representing only a small fraction of the total areas (on average 1.5 and 3.7 %, respectively). Problematic areas based on the models are Archipelago Sea, Stockholm Archipelago and Western Gulf of Finland. Those areas seem to be topographically prone to oxygen deficiency. Moreover, around 10 % of areas in Eastern Gulf of Finland are vulnerable to occasional moderate hypoxia, but less to severe hypoxia. Areas predicted as hypoxic in Gulf of Bothnia were less than < 2 %, which supports our hypothesis of the facilitating role that topography potential has. There are fewer depressions (Supporting Fig. 1), and the seafloor is topographically less complex than in the other study areas.

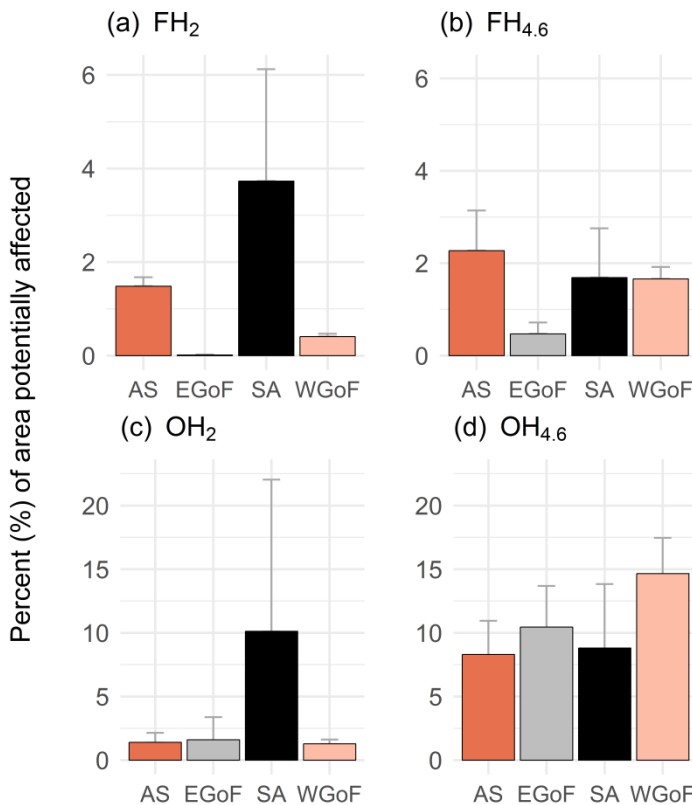

**Figure 6**. Percent (%) of areas potentially affected by hypoxia with varying frequencies (occasional and frequent) and hypoxia severities ($O_2 < 4.6$ mg L$^{-1}$ and $O_2 < 2$ mg L$^{-1}$). AS=Archipelago Sea, EGoF=Eastern Gulf of Finland, SA=Stockholm

273     Archipelago and WGoF=Western Gulf of Finland. GoB not reported as areas potentially affected were below 2 % across all

274     models.

## 4 DISCUSSION

Hypoxia has been increasing steadily since the 1960s, and anoxic areas are seizing the seafloor, suffocating marine organisms on the way (Diaz and Rosenberg, 2008;Breitburg et al., 2018). Understanding the factors affecting the severity and spatial extent of hypoxia is essential in order to estimate rates of deoxygenation and its consequences to the marine ecosystems (Breitburg et al., 2018). Earlier studies have reported coastal hypoxia to be a global phenomenon (Diaz and Rosenberg, 2008;Conley et al., 2011), and is known to be widespread in the Baltic Sea (Conley et al., 2011). Our results confirmed this, and showed that coastal hypoxia is perhaps a more common phenomenon than previously anticipated. According to our results, over 50 % of sites in the complex archipelagoes of Finland and Sweden experienced hypoxia that is ecologically significant ($O^2 < 4.6$ mg $L^{-1}$). Especially alarming was the intensity of it. For instance, Stockholm Archipelago suffered frequently from severe hypoxia ($O_2 < 2$ mg $L^{-1}$), as approximately half of the coastal monitoring sites were hypoxic across our study period (Fig. 3). This demonstrates that deoxygenated seafloors are probably even more common in coastal environments than previously reported (Karlsson et al., 2010;Conley et al., 2011). It is notable that in areas above the permanent halocline, hypoxia is in many areas seasonal, and develops after the building of thermocline in late summer (Conley et al., 2011). It is therefore probable that many of the areas we recognized as hypoxic may well be oxygenated during winter and spring. This does not however reduce the severity of the phenomenon. Even hypoxic event of short duration, e.g. few days, will reduce ecosystem resistance to further hypoxic perturbation and affect the overall ecosystem functioning (Villnas et al., 2013).

As our study suggests, topographically prone areas to deoxygenation represent less than 25 % of seascapes. However, most of the underwater nature values in the Finnish sea areas are concentrated on relatively shallow areas where there exist enough light and suitable substrates (Virtanen et al., 2018;Lappalainen et al., 2019). Shallow areas also suffer from eutrophication and rising temperatures due to changing climate, and are most probably the ones that are particularly susceptible to hypoxia in the future (Breitburg et al., 2018). This suggests that seasonal hypoxia may become a recurrent phenomenon in shallow areas above the thermocline in late summer.

Although extensive 3D models have been developed for the main basins of the Baltic Sea (Meier et al., 2011b;Meier et al., 2012a;Meier et al., 2014) the previous reports on the occurrence of coastal hypoxia have mostly been based on point observations (Conley et al., 2009a;Conley et al., 2011). Due to the lack of data, and computational limitations, no biogeochemical model has (yet) encompassed the complex Baltic Sea archipelago with a resolution needed for adjusting local management decisions. This study provides a novel methodology to predict and identify areas prone to coastal hypoxia without data on currents, stratification or biological variables, and without complex biogeochemical models. Our approach is applicable to other low-energy and non-tidal systems, such as large shallow bays and semi-enclosed or enclosed sea areas. The benefit of this approach is that it requires far less computational power than a fine-scale 3D numerical modelling. By using relatively simple proxies describing depressions of stagnant water, we were able to create detailed hypoxia maps for

the entire Finnish coastal area (23 500 km$^2$) and Stockholm Archipelago (5100 km$^2$), thus enabling a quick view of
potentially hypoxic waters (Fig. 7).

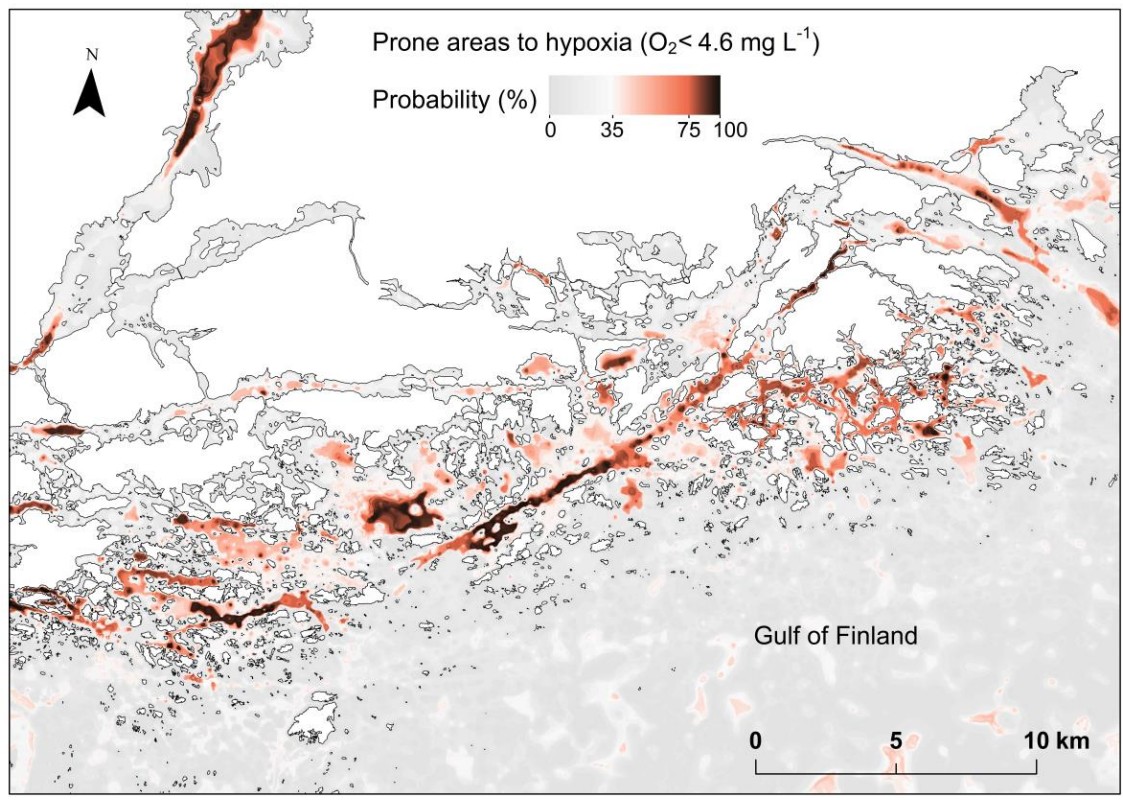

**Figure 7**. Modelled probability of detecting frequent hypoxia O$_2$ (<4.6 mg L$^{-1}$) in an example area in the south coast of
Finland. Land shown as white color.

We quantified the facilitating role of seafloor complexity for the formation of hypoxia. Sheltered, topographically
heterogeneous areas, where water exchange is limited are more susceptible for developing hypoxia according to our results.
This statement is quite intuitive, but it has not previously been quantified. Noteworthy is that coastal hypoxia is not only
related to depth; deep seafloors are not automatically hypoxic or anoxic. In our study, hypoxia was common in shallow and
moderate depths of 10–45 m. For instance in the Archipelago Sea, deep (60–100 m) channels are not usually hypoxic, as
strong currents tend to keep them oxygenated throughout the year (Virtasalo et al., 2005). Shallow areas can be restricted by
slow water movement due to topographical reasons, thus creating opportunities for hypoxia formation.
We emphasize that our models only indicate where hypoxia may occur simply due to restricted water exchange. Any
deviations from this pattern are probably caused by either hydrographic factors, which the hypoxia model based on
topography did not account for (such as strong currents in elongated, wide channels), or biogeochemical factors. Especially
high external loading, and local biogeochemical and biological processes (nutrient cycling between sediment and the water),
obviously modify the patterns and severity of hypoxia also in the coastal areas. Oxygen deficiency has been projected to
increase faster in the coastal systems than in the open sea (Gilbert et al., 2010;Altieri and Gedan, 2015). Such coastal areas
are usually affected by external nutrient loading from the watershed. In larger basins and sea areas dominated by large river
systems, such as the central Baltic Sea, Gulf of Mexico and Chesapeake Bay, large scale oceanographic and biogeochemical
processes, or external loading, govern the depth and extent of hypoxia. This is the case also in the Baltic Sea. In areas where
our models underestimate oxygen deficiency, major nutrient sources, e.g. rivers, cities or intensive agricultural areas,
probably contribute to hypoxia formation. However, in extremely complex archipelago areas, such as Finnish and Swedish
archipelago, physical factors limiting lateral and vertical movement of water probably facilitates, and in some areas even
dictates the development of hypoxia.
There were spatial differences in the frequency and severity of hypoxia that can be explained by topographical characteristics
of the areas, external loading, and interaction with the adjacent deeper basins. For instance, in the Stockholm Archipelago
severe hypoxia covered the largest percentage of seascapes of all study areas. Stockholm Archipelago is part of a joint valley
landscape with deep, steep areas also in the inner parts where wave forcing is exceptionally low, and disconnected from the
open sea, making it very susceptible for hypoxia, which was also confirmed by the model. In Finland, the inner archipelago
is mostly shallow, with steep but wider channels occurring only in the Archipelago Sea. These elongated channels are
connected to adjacent open sea areas, and thus well-ventilated, as opposite to the narrow channels of Stockholm
Archipelago. Geographically, hypoxia was in Finland most prominent in the Archipelago Sea and the Gulf of Finland, where
the inner archipelago is isolated from the open sea, and the complex topography results in overall poor water exchange in the
existing depressions. Both Stockholm Archipelago and the Archipelago Sea suffer from external loading from the associated
watersheds, and internal loading from sediments, which probably contributes to the poor oxygen status of these areas
(Puttonen et al., 2014;Walve et al., 2018). Biogeochemical factors were however not accounted for by our analysis, and
cannot be used in explaining the observed spatial differences.
In the Gulf of Finland, eutrophication increases in the open sea from west to east, which has traditionally been explained by
nutrient discharges from the Neva River (HELCOM, 2018). In our data there was however no clear gradient of coastal
hypoxia increasing towards east. In contrast, frequent hypoxia was more common in the Archipelago Sea and Western Gulf
of Finland than in the Eastern Gulf of Finland, where hypoxia occurred only occasionally. This suggests that the coastal
hypoxia is more dependent on local processes, i.e. internal loading and external loading from nearby areas, whereas open sea
hypoxia is governed by basin-scale dynamics. However, the occasional nature of hypoxia in the Eastern Gulf of Finland may
be at least partly caused by the dependency on the deep waters of the open parts of the Gulf of Finland. The Gulf of Finland
is an embayment, 400 km long and 50–120 km wide, which has an open western boundary to the Baltic Proper. A tongue of
anoxic water usually extends from the central Baltic Sea into the Gulf of Finland along its deepest parts. Basin scale
oceanographic and atmospheric processes influence how far east this tongue proceeds into the Gulf of Finland each year
(Alenius et al., 2016). It is possible that when this anoxic tongue extends close to Eastern Gulf of Finland, it also worsens the
oxygen situation of the EGoF archipelago.
In the Gulf of Bothnia, hypoxia was markedly less frequent and severe than in the other study areas. GoB has a relatively
open coastline with only few depressions (cf. Supporting Fig. 1) and strong wave forcing, which probably enhances the
mixing of water in the coastal areas. Moreover, as the open sea areas of the Gulf of Bothnia are well oxygenated due to a
lack of halocline and topographical isolation of GoB from the Baltic Proper (by the sill between these basins) (Leppäranta
and Myrberg, 2009), hypoxic water is not advected from the open sea to the coastal areas.
Such observations suggest that formation of coastal hypoxia is not totally independent from basin-scale oxygen dynamics.
While we suggest that coastal hypoxia can be formed entirely based on local morphology and local biogeochemical
processes, the relatively low occurrence of hypoxia in the Gulf of Bothnia, and differences in frequency of hypoxia in
different parts of the Gulf of Finland, both highlight the interaction of these coastal areas with the Baltic Proper.
While our results confirm that hypoxia in most study areas is a frequently occurring phenomenon, they also show that areas
affected by hypoxia are geographically still limited. Our modelling results indicate that overall, less than 25 % of the studied
sea areas were afflicted by some form of hypoxia (be it recurrent or occasional), and less than 6 % of seascapes were plagued
by frequent, severe hypoxia. The relatively small spatial extent of coastal hypoxia does not mean that it is not a harmful
phenomenon. In Stockholm Archipelago, severe hypoxia is a pervasive and persistent phenomenon, and also in Finland,
many local depressions are often hypoxic. Anoxic local depressions probably act as local nutrient sources, releasing
especially phosphorus to the water column, which further enhances pelagic primary production. Such a vicious circle tends
to worsen the eutrophication and maintain the environment in a poor state (Pitkänen et al., 2001;Vahtera et al., 2007). In this
way, even small sized anoxic depressions, especially if they are many, may affect the ecological status of the whole coastal
area. Moreover, as climate change has been projected to increase water temperatures and worsen hypoxia in the Baltic Sea
(Meier et al., 2011a), shallow archipelago areas that typically have high productivity, warm up quickly, and are
topographically prone to hypoxia, may be especially vulnerable to the negative effects of climate change.
In order to establish reference conditions and implement necessary and cost-efficient measures to reach the goals of
international agreements such as the EU Water Framework Directive (WFD, 2000/60/EC), the Marine Strategy Framework
Directive (MSFD, 2008/56/EC) and the HELCOM Baltic Sea Action Plan (BSAP), in-depth knowledge of ecological
functions and processes as well as natural preconditions is needed. Although eutrophication is a problem for the whole Baltic
Sea, nutrient abatement measures are taken locally. We therefore need to know where the environmental benefits are
maximized, and where natural conditions are likely to counteract any measures taken. As some places are naturally prone to
hypoxia, our model could aid directing measures to places where they are most likely to be efficient, as well as explain why
in some areas implemented measures do not have the desired effect. Our approach could be used to develop an "early-
warning system" for identification of areas prone to oxygen loss, and to indicate where eutrophication mitigation actions are
most urgently needed.

## CONCLUSIONS

While biogeochemical 3D models have been able to accurately project basin-scale oxygen dynamics, describing spatial variation of hypoxia in coastal areas has remained a challenge. Recognizing that the enclosed nature of seafloors contributes to hypoxia formation, we used simple topographical parameters to model the occurrence of hypoxia in the complex Finnish and Swedish archipelagoes. We found that a surprisingly large fraction (~80 %) of hypoxia occurrences could be explained by topographical parameters alone. Modelling results also suggested that less than 25 % of the studied seascapes were prone to hypoxia during late summer. Large variation existed in the spatial and temporal patterns of hypoxia, however, with certain areas being prone to occasional severe hypoxia ($O_2 < 2$ mg/L), while others were more susceptible to recurrent moderate hypoxia ($O_2 < 4.6$ mg/L). Sheltered, topographically heterogeneous areas with limited water exchange were susceptible for developing hypoxia, in contrast to less sheltered areas with high wave forcing. In some areas oxygen conditions were either better or worse than predicted by the model. We assume that these deviations from the "topographical background" were caused by processes not accounted for by the model, such as hydrographical processes, e.g. strong currents causing improved mixing, or by high external or internal nutrient loading, inducing high local oxygen consumption. We conclude that formation of coastal hypoxia is probably primarily dictated by local processes, and can be quite accurately projected using simple topographical parameters, but that interaction with the associated watershed and the adjacent deeper basins of the Baltic Sea can also influence local oxygen dynamics in many areas. Our approach gives a practical baseline for various types of hypoxia related studies and consequently, decision-making. Identifying areas prone to hypoxia helps to focus research, management and conservation actions in a cost-effective way.

## AUTHOR CONTRIBUTION

EAV and MV designed the study, EAV and ANS performed all analyses, EAV wrote the main text and all authors contributed to the writing and editing of the manuscript.

## DATA AVAILABILITY

Data and model codes will be submitted to Dryad data repository.

## ACKNOWLEDGMENTS

E.A.V and M.V. acknowledge the SmartSea project (Grant nos. 292985 and 314225), funded by the Strategic Research Council of the Academy of Finland, and the Finnish Inventory Programme for the Underwater Marine Environment VELMU, funded by the Ministry of the Environment. A.N acknowledges the support of the Academy of Finland

(project 294853) and the Sophie von Julins Foundation. A.N.S. acknowledges the IMAGINE project (Grant no 15/247),
funded by the Swedish EPA.

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
