# Peer review of "Identifying areas prone to coastal hypoxia - the role of topography"

_Biogeosciences, 2019_

## Referee Comment (RC1) · Anonymous Referee #1 · 13 May 2019

**General comments**

This study presents a GIS analysis of the ability of selected geomorphological metrics to predict the occurrence of seafloor hypoxia in the coastal sea areas of Finland and Stockholm Archipelago in the northern Baltic Sea. The selection of geomorphological metrics is well justified, and the predictive performance of the metrics is evaluated against subsets of national water quality monitoring data from 808 sites. The selected metrics have been described and published elsewhere, but used in this study in a combination that is itself novel. The key results of the analysis are that Depth-Attenuated Wave Exposure (SWM(d)) is the most important metric, and that all the metrics combined predict hypoxia correctly in >80% of the cases. The main conclusion that sluggish water exchange increases the development of hypoxia in patchy archipelago areas is

not particularly new as correctly pointed out by the authors, but this study nevertheless contributes to the understanding of mechanisms driving hypoxia, and the significance of topography in particular. In general, the manuscript is well written and illustrated, and the topic is suitable for Biogeosciences. However, as usual, there is also room for improvement. Overall, I recommend this study for publication after my comments below have been adequately addressed.

This manuscript is focused on topography, while the other known drivers of hypoxia are less considered, including the other physical drivers. The coastal areas studied in the manuscript generally lie above the Baltic Sea halocline, where the seasonal development of thermocline is an important feature with respect to reduced water exchange and hypoxia. The seasonality of hypoxia is mentioned in the manuscript, but it needs to be emphasized more that the implied coastal hypoxia at large is seasonal by nature, and different from the more permanent hypoxia in areas below halocline. The authors may even consider including the word "seasonal" in the title. The authors further need to discuss the potential effects thermocline has on hypoxia in the shallow sea areas. The authors may even consider exploring, whether the typical depth of thermocline could be included in the analysis in order to improve the predictive ability of metrics.

An important conclusion is that half of the monitoring sites in Stockholm Archipelago and one third of sites in southern Finland experienced severe hypoxia. It should be discussed whether this is an artefact resulting from the locations of monitoring sites or a true difference between these two sea areas.

The authors conclude (e.g. page 15) that hypoxia most often occurs in shallow to moderate water depths, in accordance with previous studies. However, looking at Figure 7, hypoxia seems to be developed in deep channels, which in the previous studies have been concluded to be well ventilated. This discrepancy needs to be discussed further.

Terminology used in the manuscript is partly confusing, and the authors may want to seek help from a colleague with background in seafloor geology/sedimentology in
particular. The term "sinkhole" is widely used in the manuscript, although collapse structures are unlikely in the study area with predominantly siliciclastic sediments.

Specific comments

Abstract, line 12. Add "and vertical mixing" between "water circulation" and "that can".

Abstract, line 17. Replace "sinkholes" by "local depressions".

Abstract, line 17. Add "seasonal" between "development of" and "hypoxia".

Page 3, line 21. Add "Sea" between "Baltic" and "coastal". Here and elsewhere in the manuscript, note that the term Baltic when used alone refers to the Baltic States.

Page 4, line 1. Replace "archipelago" with "islands". There is no archipelago really in GoB (except Vaasa).

Page 4, lines 3-4. Specify the type of soft sediments in shallow areas. Organic-rich mud?

Page 4, line 4. Replace "rocky" with "hard clay, till and bedrock". Rocky is an oversimplification.

Page 10, line 10. Replace "Contrary to" with "In contrast in".

Page 11, line 7, and elsewhere in the manuscript. The use of word "sink" is very confusing in this context and should be replaced by a more correct term.

Page 14, line 6. The number "20 %" probably does not make much sense here, because the more sites one would sample in the Baltic Sea, the higher would be the number of hypoxic sites. It is probably sufficient to state that hypoxia is known to be widespread in the Baltic Sea.

Page 16, line 10. Replace "canyons" with "channels".

Page 16, line 16, and elsewhere in the manuscript. Replace "ledge" with "tongue".

BGD
Page 16, line 17-19. What would be the contribution of River Neva to the hydrodynamics and hypoxia in the EGoF?

Page 16, lines 22-26. In case the authors insist that the lack coastal hypoxia in GoB is due to the lack of halocline and permanent hypoxia in central deep areas, the driving mechanisms need to be explained. Probably it is safer to just state that there is less hypoxia in GoB coastal areas because of less islands and stronger wave forcing.

Page 16, lines 27-30. If valid, this conclusion needs to be better substantiated. The authors write many times in the manuscript that the deepest parts (channels) in archipelago areas are usually well oxygenated. How does that oxic deep water then transform to shallow water hypoxia?

The Conclusions section as it is currently written is more about the implications of findings than the actual conclusions of the study.

The electronic supplement to this manuscript only has one figure. This figure is quite informative, and the authors may wish to consider including it as a figure in the actual paper.

That ends my referee comments.

BGD

---

## Referee Comment (RC2) · Anonymous Referee #2 · 20 Jun 2019

Thank you for inviting me to review "Identifying areas prone to coastal hypoxia -the role of topography" by Virtanen et al, submitted to BIogeosciences. In general, this is a very well written, well-structured and interesting study where the authors use new approaches to quantify the impact of topography on bottom-water [O2]. I strongly recommend it to be published in Biogeosciences and I hope the approaches presented will be widely used for also other geographical areas. My major objection is the following statements in the beginning of the manuscripts: "We hypothesized that the enclosed nature of seafloors facilitates hypoxia formation." "We discovered that topographically sheltered seafloors and sinkholes with stagnant water are prone to the development of hypoxia" It is text book knowledge that topography (i.e. sills, deep basins, restricted morphology, skerries etc.) has a large impact on residence time and water circulation,

hence also on dissolved [O2] in the bottom water. I honestly don't think this was new knowledge for the authors and hence the main driver of the study. However, and here is where the study becomes more interesting, to determine the degree to which a restricted setting affects the [O2] (i.e. the quantification) and then model that effect. That is interesting and new. I would like to see the author rephrasing their aim and their hypothesis.

Minor details: It would help the reader to use abbreviations as sparingly as possible and to remind us what the geographical abbreviations stand for in the beginning of the result/discussion section, and preferable use the names of the regions more in the text. I find it slightly difficult to accept the term normoxic and that is defined as > 4.6 mg/l. What is the "normal"/norm for a deep-water in a coastal setting, should we expect fully oxygenated conditions, should that our reference value? It is important to think about in these type of studies. Dead zones is a popular science word which isn't really accurate, dead zones are devoid of higher life but not of all organisms. It should be used within " ", if used at all in this type of publications. The conclusions are very short, general and undersell the study. I would suggest the authors to be more detailed and really highlight the specific conclusions from the study. One of them is: topographically prone areas to deoxygenation represent less than 25 % of the investigated seascapes.

The link to SMHI doesn't work (paragraph 2.2). The references are not consitently formatted.

I can't evaluate the modelling approaches, as that is far from my field, and I hope a second reviewer can do that.

I'm looking forward to see the study published.

---

## Author Comment (AC1) · 24 Jun 2019

We thank Referee #2 for insightful comments on our manuscript that improved it substantially. We have now taken into account all the comments received and edited text accordingly. Referee #2 comments marked as grey and our responses as black.

Thank you for inviting me to review "Identifying areas prone to coastal hypoxia -the role of topography" by Virtanen et al, submitted to BIogeosciences. In general, this is a very well written, well-structured and interesting study where the authors use new approaches to quantify the impact of topography on bottom-water [O2]. I strongly recommend it to be published in Biogeosciences and I hope the approaches presented will be widely used for also other geographical areas. My major objection is the following statements in the beginning of the manuscripts: "We hypothesized that the enclosed nature of seafloors facilitates hypoxia formation." "We discovered that topographically sheltered seafloors and sinkholes with stagnant water are prone to the development of hypoxia" It is text book knowledge that topography (i.e. sills, deep basins, restricted morphology, skerries etc.) has a large impact on residence time and water circulation, hence also on dissolved [O2] in the bottom water. I honestly don't think this was new knowledge for the authors and hence the main driver of the study. However, and here is where the study becomes more interesting, to determine the degree to which a restricted setting affects the [O2] (i.e. the quantification) and then model that effect. That is interesting and new. I would like to see the author rephrasing their aim and their hypothesis.
We have now edited the aims and hypothesis of our manuscript, and changed texts in Abstract and in Introduction.

In Abstract:
"It is well known that the enclosed nature of seafloors and reduced water mixing facilitates hypoxia formation, but the degree to which topography contributes to hypoxia formation, and small-scale variability of coastal hypoxia, has not been previously quantified."

And:
"We developed simple proxies of seafloor heterogeneity and modelled oxygen deficiency in complex coastal areas in the northern Baltic Sea. According to our models, topographical parameters alone explained ~80 % of hypoxia occurrences. The models also revealed that less than 25 % of the studied seascapes were prone to hypoxia during late summer (August-September)."

In Introduction:

"…It is widely recognized that the semi-enclosed nature of the seafloors, and associated limited water exchange is a significant factor in the formation of hypoxia in coastal waters (Rabalais et al., 2010;Conley et al., 2011;Diaz and Rosenberg, 1995a;Virtasalo et al., 2005). However, to determine the degree to which seascape structure restricting water movement, contributes to hypoxia formation has not been quantified. …"

And:
"We tested how large fraction of hypoxia occurrences could be explained only by structural complexity of seascapes, without knowledge on hydrographical or biogeochemical parameters."

Minor details: It would help the reader to use abbreviations as sparingly as possible and to remind us what the geographical abbreviations stand for in the beginning of the result/discussion section, and preferable use the names of the regions more in the text.
Abbreviations replaced with place names accordingly in the text to help the reader, both in the results section and in the discussion.

I find it slightly difficult to accept the term normoxic and that is defined as > 4.6 mg/l. What is the "normal"/norm for a deep-water in a coastal setting, should we expect fully oxygenated conditions, should that our reference value? It is important to think about in these type of studies.
We agree that "normal oxygen conditions" are difficult to define in an environment like the Baltic Sea.

We have deleted the term "normoxic" and edited text accordingly:

"…to discriminate a hypoxic site from a **normoxic** one" changed to:
"…to discriminate a hypoxic site from **an oxic one**"

"…AS, EGoF and SA were **normoxic**…" changed to:
"…AS, EGoF and SA were **not hypoxic**…"

"…channels are mostly **normoxic**…" changed to:
"…channels are not **usually hypoxic**…"

"…many local depressions are more often hypoxic than **normoxic**…" changed to:
"…many local depressions are **often hypoxic**…"

The threshold for hypoxia admittedly varies in literature, but here we define it as O2 >4.6 mg/l. This limit is mentioned Section 2.2. Hypoxia data:

"Here we define hypoxia based on two ecologically meaningful limits: moderately hypoxic <4.6 mg L -1 O2 – as this has been estimated to be a minimum safe limit for species survival, behavior and functioning in benthic communities (Norkko et al., 2015)"

Dead zones is a popular science word which isn't really accurate, dead zones are devoid of higher life but not of all organisms. It should be used within " ", if used at all in this type of publications.
Dead zones deleted from the text, and now we talk about anoxic areas and anoxic zoned devoid of higher life throughout the text.

The conclusions are very short, general and undersell the study. I would suggest the authors to be more detailed and really highlight the specific conclusions from the study. One of them is: topographically prone areas to deoxygenation represent less than 25 % of the investigated seascapes.
Conclusions (and Abstract) are now thoroughly edited to highlight the key findings of the study:

"We found that a surprisingly large fraction (~80 %) of hypoxia occurrences could be explained by topographical parameters alone. Modelling results also suggested that less than 25 % of the studied seascapes were prone to hypoxia during late summer. Large variation existed in the spatial and temporal patterns of hypoxia, however, with certain areas being prone to occasional severe hypoxia (O2 < 2 mg/L), while others were more susceptible to recurrent moderate hypoxia (O2 < 4.6 mg/L). Sheltered, topographically heterogeneous areas with limited water exchange were susceptible for developing hypoxia, in contrast to less sheltered areas with high wave forcing. In some areas oxygen conditions were either better or worse than predicted by the model. We assume that these deviations from the "topographical background" were caused by processes not accounted for by the model, such as hydrographical processes, e.g. strong currents causing improved mixing, or by high external or internal nutrient loading, inducing high local oxygen consumption. We conclude that formation of coastal hypoxia is probably primarily dictated by local processes, and can be quite accurately projected using simple topographical parameters, but that interaction with the associated watershed and the adjacent deeper basins of the Baltic Sea can also influence local oxygen dynamics in many areas.."

The link to SMHI doesn't work (paragraph 2.2).
Hyperlink changed to a link that works: https://www.smhi.se/data/oceanografi/havsmiljodata

The references are not consistently formatted.

References checked and reformatted.

I can't evaluate the modelling approaches, as that is far from my field, and I hope a second reviewer can do that.

I'm looking forward to see the study published.

---

## Author Comment (AC2) · 24 Jun 2019

We wish to thank Anonymous Referee #1 for insightful ideas and comments, they improved the manuscript substantially. Our comments marked as black, Referee #1´s as grey.

**General comments**

This study presents a GIS analysis of the ability of selected geomorphological metrics to predict the occurrence of seafloor hypoxia in the coastal sea areas of Finland and Stockholm Archipelago in the northern Baltic Sea. The selection of geomorphological metrics is well justified, and the predictive performance of the metrics is evaluated against subsets of national water quality monitoring data from 808 sites. The selected metrics have been described and published elsewhere, but used in this study in a combination that is itself novel. The key results of the analysis are that Depth-Attenuated Wave Exposure (SWM(d)) is the most important metric, and that all the metrics combined predict hypoxia correctly in >80% of the cases. The main conclusion that sluggish water exchange increases the development of hypoxia in patchy archipelago areas is not particularly new as correctly pointed out by the authors, but this study nevertheless contributes to the understanding of mechanisms driving hypoxia, and the significance of topography in particular. In general, the manuscript is well written and illustrated, and the topic is suitable for Biogeosciences. However, as usual, there is also room for improvement. Overall, I recommend this study for publication after my comments below have been adequately addressed.

This manuscript is focused on topography, while the other known drivers of hypoxia are less considered, including the other physical drivers. The coastal areas studied in the manuscript generally lie above the Baltic Sea halocline, where the seasonal development of thermocline is an important feature with respect to reduced water exchange and hypoxia. The seasonality of hypoxia is mentioned in the manuscript, but it needs to be emphasized more that the implied coastal hypoxia at large is seasonal by nature, and different from the more permanent hypoxia in areas below halocline. The authors may even consider including the word "seasonal" in the title.

Good point. We recognize that hypoxia is in many shallow areas above halocline seasonal. We have added text about this in Discussion:

> "It is notable that in areas above the permanent halocline, hypoxia is in many areas seasonal, and develops after the building of thermocline in late summer (Conley et al., 2011). It is therefore probable that many of the areas we recognized as hypoxic may well be oxygenated during winter and spring. This does not however reduce the severity of the phenomenon. Even hypoxic event of short duration, e.g. few days, will reduce ecosystem resistance to further hypoxic perturbation and affect the overall ecosystem functioning (Villnas et al., 2013)."

The authors further need to discuss the potential effects thermocline has on hypoxia in the shallow sea areas.

We also added a short notion of warming up and thermocline in Discussion, where potential effects of climate change are discussed:

> "Shallow areas also suffer from eutrophication and rising temperatures due to changing climate, and are most probably the ones that are particularly susceptible to hypoxia in the future (Breitburg et al., 2018). This suggests that seasonal hypoxia may become a recurrent phenomenon in shallow areas above the thermocline in late summer."

The authors may even consider exploring, whether the typical depth of thermocline could be included in the analysis in order to improve the predictive ability of metrics.

The suggestion that the depth of thermocline could be included in the analysis is an interesting idea, and could be explored in further development of the model. However, our modelling approach concentrates on the role topography has, and how much of the hypoxia occurrence and variability can be explained by topographical parameters alone. It can well be that adding information about the thermocline depth could improve the model performance (although already decent: ~80 % of the hypoxia occurrences explained

against independent data). Moreover, thermocline depth can be highly variable between 1–10m, and data for defining that for prediction is not so easy to obtain. We have however included depth as an explanatory factor in our model. The analysis showed that depth was the second most important factor in explaining the variation in hypoxia (after depth-attenuated exposure; cf. Fig. 4). We therefore do not consider it possible to add thermocline *per se* in the analysis for now with the data limitations.

We have however clarified the interaction of depth with sheltered areas in Results:

> "Noteworthy is also that depth was not the most important driver of hypoxia in coastal areas. This suggests that hypoxia is not directly dependent on depth, but that depressions that are especially steep and isolated are more sheltered and become more easily hypoxic than smoother depressions."

An important conclusion is that half of the monitoring sites in Stockholm Archipelago and one third of sites in southern Finland experienced severe hypoxia. It should be discussed whether this is an artefact resulting from the locations of monitoring sites or a true difference between these two sea areas.

A good point. We do not find any major differences in sampling strategy or methods, and do not see any reason to doubt the validity of this result. Stockholm Archipelago is characterised by a particularly steep topography, with narrow channels, where wave forcing is extremely low, making it thus very susceptible for hypoxia to develop (and confirmed by the model). This has been now further explained in text:

> "There were spatial differences in the frequency and severity of hypoxia that can be explained by topographical characteristics of the areas, external loading, and interaction with the adjacent deeper basins. For instance, in the Stockholm Archipelago severe hypoxia covered the largest percentage of seascapes of all study areas. Stockholm Archipelago is part of a joint valley landscape with deep, steep areas also in the inner parts where wave forcing is exceptionally low, making it very susceptible for hypoxia, which was also confirmed by the model. In Finland, the inner archipelago is mostly shallow, with steep but wider channels occurring only in the Archipelago Sea. Geographically, hypoxia was in Finland most prominent in the Archipelago Sea and the Gulf of Finland, where the inner archipelago is isolated from the open sea, and the complex topography results in overall poor water exchange in the existing depressions."

We also added in the end of the same paragraph text on the possible role of external nutrient loading:

> "Both Stockholm Archipelago and the Archipelago Sea suffer from external loading from the associated watersheds, and internal loading from sediments, which probably contributes to the poor oxygen status of these areas (Walve et al., 2018;Puttonen et al., 2014). Biogeochemical factors were however not accounted for by our analysis, and cannot be used in explaining the observed spatial differences."

The authors conclude (e.g. page 15) that hypoxia most often occurs in shallow to moderate water depths, in accordance with previous studies. However, looking at Figure 7, hypoxia seems to be developed in deep channels, which in the previous studies have been concluded to be well ventilated. This discrepancy needs to be discussed further.

Actually, the elongated, narrow channel in Fig. 7 is not very deep (mean depth 20 m), and it is not very well ventilated because it is not connected to open areas. In contrast, the channels in the Archipelago Sea are much wider and deeper, and well connected to the open sea, and thus experiencing higher water mixing. The reviewer however raises an important point, and we thank the referee for raising this issue. We emphasize that our study cannot fully take into account hydrographic factors, such as strong currents in deep, elongated channels (such as the ones crisscrossing the Archipelago Sea). We merely aim to indicate areas that are topographically susceptible to hypoxia. For instance, the deep, narrow channels of the Stockholm Archipelago are more susceptible for hypoxia formation than wider, long channels of the Archipelago Sea with connectivity to adjacent open sea areas. We have now emphasized this point even further throughout the manuscript:

In Abstract:
> "It is well known that the enclosed nature of seafloors and reduced water mixing facilitates hypoxia formation, but the degree to which topography contributes to hypoxia formation, and small-scale variability of coastal hypoxia, has not been previously quantified. We developed simple proxies of seafloor heterogeneity and modelled oxygen deficiency in

complex coastal areas in the northern Baltic Sea. According to our models, topographical parameters alone explained ~80 % of hypoxia occurrences."

"…Deviations from this "topographical background" are probably caused by strong currents or by high nutrient loading, thus improving or worsening oxygen status, respectively. In some areas, connectivity with adjacent deeper basins may also influence coastal oxygen dynamics.".

In Introduction:

"It is widely recognized that the semi-enclosed nature of the seafloors, and associated limited water exchange is a significant factor in the formation of hypoxia in coastal waters (Rabalais et al., 2010;Conley et al., 2011;Diaz and Rosenberg, 1995a;Virtasalo et al., 2005). However, to determine the degree to which seascape structure restricting water movement contributes to hypoxia formation has not been quantified."

"…We tested how large fraction of hypoxia occurrences could be explained only by structural complexity of seascapes, without knowledge on hydrographical or biogeochemical parameters."

In Discussion:

"We emphasize that our models only indicate where hypoxia may occur simply due to restricted water exchange. Any deviations from this pattern are probably caused by either hydrographic factors, which the hypoxia model based on topography did not account for (such as strong currents in elongated, wide channels), or biogeochemical factors. Especially high external loading, and local biogeochemical and biological processes (nutrient cycling between sediment and the water), obviously modify the patterns and severity of hypoxia also in the coastal areas"

Terminology used in the manuscript is partly confusing, and the authors may want to seek help from a colleague with background in seafloor geology/sedimentology in particular. The term "sinkhole" is widely used in the manuscript, although collapse structures are unlikely in the study area with predominantly siliciclastic sediments.

We acknowledge that "sinkhole" is a confusing term. We´ve changed that to "local depressions" were appropriate.

**Specific comments**

Abstract, line 12. Add "and vertical mixing" between "water circulation" and "that can".

Added "vertical mixing".

Abstract, line 17. Replace "sinkholes" by "local depressions".

Sinkholes replaced with "local depressions"

Abstract, line 17. Add "seasonal" between "development of" and "hypoxia".

Abstract reformatted thoroughly, and we speak now of hypoxia during late summer (August-September):

"Hypoxia is an increasing problem in marine ecosystems around the world. While major advances have been made in our understanding of the drivers of hypoxia, challenges remain in describing oxygen dynamics in coastal regions. The complexity of many coastal areas and lack of detailed *in situ* data has hindered the development of models describing oxygen dynamics at a sufficient spatial resolution for efficient management actions to take place. It is well known that the enclosed nature of seafloors and reduced water mixing facilitates hypoxia formation, but the degree to which topography contributes to hypoxia formation, and small-scale variability of coastal hypoxia, has not been previously quantified. We developed simple proxies of seafloor heterogeneity and modelled oxygen deficiency in complex coastal areas in the northern Baltic Sea. According to our models, topographical parameters alone explained ~80 % of hypoxia occurrences. The models also revealed that less than 25 % of the studied seascapes were prone to hypoxia during late summer (August-September). However, large variation existed in the spatial and temporal patterns of hypoxia, as certain areas were prone to occasional severe hypoxia ($O_2 < 2$ mg L$^{-1}$), while others were more susceptible to recurrent moderate hypoxia ($O2 < 4.6$ mg L$^{-1}$). Areas identified as problematic in our study were characterized by low exposure to wave forcing, by high topographical shelter from surrounding areas, and by isolation from the open sea, all contributing to longer water residence times in seabed depressions. Deviations from this "topographical background" are probably caused by strong currents or by high nutrient loading, thus improving or worsening oxygen status, respectively. In some areas, connectivity with adjacent deeper basins may also influence coastal oxygen dynamics. Developed models could boost the performance of biogeochemical models, aid developing nutrient abatement measures, and pinpoint areas where management actions are most urgently needed."

Page 3, line 21. Add "Sea" between "Baltic" and "coastal". Here and elsewhere in the manuscript, note that the term Baltic when used alone refers to the Baltic States.
 "Sea" added.

Page 4, line 1. Replace "archipelago" with "islands". There is no archipelago really in GoB (except Vaasa).
Changed to "islands"

Page 4, lines 3-4. Specify the type of soft sediments in shallow areas. Organic-rich mud?
Sentence reformatted to:

> "Substrate in both areas varies from organic-rich soft sediments in sheltered locations to hard clay, till and bedrock in exposed areas."

Page 4, line 4. Replace "rocky" with "hard clay, till and bedrock". Rocky is an oversimplification.
Changed to "hard clay, till and bedrock"

Page 10, line 10. Replace "Contrary to" with "In contrast in".
Replaced with "In contrast in".

Page 11, line 7, and elsewhere in the manuscript. The use of word "sink" is very confusing in this context and should be replaced by a more correct term.
Sinkholes replaced with "local depressions" throughout the manuscript.

Page 14, line 6. The number "20 %" probably does not make much sense here, because the more sites one would sample in the Baltic Sea, the higher would be the number of hypoxic sites. It is probably sufficient to state that hypoxia is known to be widespread in the Baltic Sea.

Reformatted to:

> "Earlier studies have reported coastal hypoxia to be a global phenomenon (Diaz and Rosenberg, 2008;Conley et al., 2011), and is known to be widespread in the Baltic Sea (Conley et al., 2011)."

Page 16, line 10. Replace "canyons" with "channels".
Canyons replaced with channels.

Page 16, line 16, and elsewhere in the manuscript. Replace "ledge" with "tongue".
Replaced ledge with tongue.

Page 16, line 17-19. What would be the contribution of River Neva to the hydrodynamics and hypoxia in the EGoF?
Interesting point. Reformatted to:

> "In the Gulf of Finland, eutrophication increases in the open sea from west to east, which has traditionally been explained by nutrient discharges from the Neva River (HELCOM, 2018). In our data there was however no clear gradient of coastal hypoxia increasing towards east. In contrast, frequent hypoxia was more common in the Archipelago Sea and Western Gulf of Finland than in the Eastern Gulf of Finland, where hypoxia occurred only occasionally. This suggests that the coastal hypoxia is more dependent on local processes, i.e. internal loading and external loading from nearby areas, whereas open sea hypoxia is governed by basin-scale dynamics. However, the occasional nature of hypoxia in the Eastern Gulf of Finland may be at least partly caused by the dependency on the deep waters of the open parts of the Gulf of Finland. The Gulf of Finland is an embayment, 400 km long and 50–120 km wide, which has an open western boundary to the Baltic Proper. A tongue of anoxic water usually extends from the central Baltic Sea into the Gulf of Finland along its deepest parts. Basin scale oceanographic and atmospheric processes influence how far east this tongue proceeds into the Gulf of Finland each year (Alenius et al., 2016). It is possible that when this anoxic tongue extends close to Eastern Gulf of Finland, it also worsens the oxygen situation of the EGoF archipelago."

Page 16, lines 22-26. In case the authors insist that the lack coastal hypoxia in GoB is due to the lack of halocline and permanent hypoxia in central deep areas, the driving mechanisms need to be explained.

Probably it is safer to just state that there is less hypoxia in GoB coastal areas because of less islands and stronger wave forcing.

We have clarified this:

"In the Gulf of Bothnia, hypoxia was markedly less frequent and severe than in the other study areas. GoB has a relatively open coastline with only few depressions (cf. Supporting Fig. 1) and strong wave forcing, which probably enhances the mixing of water in the coastal areas. Moreover, as the open sea areas of the Gulf of Bothnia are well oxygenated due to a lack of halocline and topographical isolation of GoB from the Baltic Proper (by the sill between these basins) (Leppäranta and Myrberg, 2009), hypoxic water is not advected from the open sea to the coastal areas."

Page 16, lines 27-30. If valid, this conclusion needs to be better substantiated. The authors write many times in the manuscript that the deepest parts (channels) in archipelago areas are usually well oxygenated. How does that oxic deep water then transform to shallow water hypoxia?

The deep elongated, but wide channels are well-ventilated in the Archipelago Sea due to the connectivity to adjacent open areas, and potentially due to strong currents, as opposite to the narrow channels of Stockholm Archipelago, which occur in the inner parts of the archipelago. When we refer to "the deeper basins" we refer to the Baltic Proper and deep parts of the Gulf of Finland. This is now clarified in the text:

"There were spatial differences in the frequency and severity of hypoxia that can be explained by topographical characteristics of the areas, external loading, and interaction with the adjacent deeper basins. For instance, in the Stockholm Archipelago severe hypoxia covered the largest percentage of seascapes of all study areas. Stockholm Archipelago is part of a joint valley landscape with deep, steep areas also in the inner parts where wave forcing is exceptionally low, and disconnected from the open sea, making it very susceptible for hypoxia, which was also confirmed by the model. In Finland, the inner archipelago is mostly shallow, with steep but wider channels occurring only in the Archipelago Sea. These elongated channels are connected to adjacent open sea areas, and thus well-ventilated, as opposite to the narrow channels of Stockholm Archipelago. Geographically, hypoxia was in Finland most prominent in the Archipelago Sea and the Gulf of Finland, where the inner archipelago is isolated from the open sea, and the complex topography results in overall poor water exchange in the existing depressions."

And:

"…While we suggest that coastal hypoxia can be formed entirely based on local morphology and local biogeochemical processes, the relatively low occurrence of hypoxia in the Gulf of Bothnia, and differences in frequency of hypoxia in different parts of the Gulf of Finland, both highlight the interaction of these coastal areas with the Baltic Proper."

The Conclusions section as it is currently written is more about the implications of findings than the actual conclusions of the study.

Noted. More findings added to the conclusions:

"While biogeochemical 3D models have been able to accurately project basin-scale oxygen dynamics, describing spatial variation of hypoxia in coastal areas has remained a challenge. Recognizing that the enclosed nature of seafloors contributes to hypoxia formation, we used simple topographical parameters to model the occurrence of hypoxia in the complex Finnish and Swedish archipelagoes. We found that a surprisingly large fraction (~80 %) of hypoxia occurrences could be explained by topographical parameters alone. Modelling results also suggested that less than 25 % of the studied seascapes were prone to hypoxia during late summer. Large variation existed in the spatial and temporal patterns of hypoxia, however, with certain areas being prone to occasional severe hypoxia ($O_2 < 2$ mg/L), while others were more susceptible to recurrent moderate hypoxia ($O_2 < 4.6$ mg/L). Sheltered, topographically heterogeneous areas with limited water exchange were susceptible for developing hypoxia, in contrast to less sheltered areas with high wave forcing. In some areas oxygen conditions were either better or worse than predicted by the model. We assume that these deviations from the "topographical background" were caused by processes not accounted for by the model, such as hydrographical processes, e.g. strong currents causing improved mixing, or by high external or internal nutrient loading, inducing high local oxygen consumption. We conclude that formation of coastal hypoxia is probably primarily dictated by local processes, and can be quite accurately projected using simple topographical parameters, but that interaction with the associated watershed and the adjacent deeper basins of the Baltic Sea can also influence local oxygen dynamics in many areas. Our approach gives a practical baseline for various types of hypoxia related studies and consequently, decision-making. Identifying areas prone to hypoxia helps to focus research, management and conservation actions in a cost-effective way."

The electronic supplement to this manuscript only has one figure. This figure is quite informative, and the authors may wish to consider including it as a figure in the actual paper.

We only refer to this figure when speaking of BPI differences in GoB and other areas, and as such we would prefer to keep it in the Supplementary material.

That ends my referee comments.